# Learning with Noisy Labels

**Nagarajan Natarajan**      **Inderjit S. Dhillon**      **Pradeep Ravikumar**
Department of Computer Science, University of Texas, Austin.
`{naga86,inderjit,pradeepr}@cs.utexas.edu`

**Ambuj Tewari**
Department of Statistics, University of Michigan, Ann Arbor.
`tewaria@umich.edu`

## Abstract

In this paper, we theoretically study the problem of binary classification in the presence of random classification noise — the learner, instead of seeing the true labels, sees labels that have independently been flipped with some small probability. Moreover, random label noise is *class-conditional* — the flip probability depends on the class. We provide two approaches to suitably modify any given surrogate loss function. First, we provide a simple unbiased estimator of any loss, and obtain performance bounds for empirical risk minimization in the presence of iid data with noisy labels. If the loss function satisfies a simple symmetry condition, we show that the method leads to an efficient algorithm for empirical minimization. Second, by leveraging a reduction of risk minimization under noisy labels to classification with weighted 0-1 loss, we suggest the use of a simple weighted surrogate loss, for which we are able to obtain strong empirical risk bounds. This approach has a very remarkable consequence — methods used in practice such as biased SVM and weighted logistic regression are provably noise-tolerant. On a synthetic non-separable dataset, our methods achieve over 88% accuracy even when 40% of the labels are corrupted, and are competitive with respect to recently proposed methods for dealing with label noise in several benchmark datasets.

## 1   Introduction

Designing supervised learning algorithms that can learn from data sets with noisy labels is a problem of great practical importance. Here, by noisy labels, we refer to the setting where an adversary has deliberately corrupted the labels [Biggio et al., 2011], which otherwise arise from some "clean" distribution; learning from only positive and unlabeled data [Elkan and Noto, 2008] can also be cast in this setting. Given the importance of learning from such noisy labels, a great deal of practical work has been done on the problem (see, for instance, the survey article by Nettleton et al. [2010]). The theoretical machine learning community has also investigated the problem of learning from noisy labels. Soon after the introduction of the noise-free PAC model, Angluin and Laird [1988] proposed the *random classification noise* (RCN) model where each label is flipped independently with some probability $\rho \in [0, 1/2)$. It is known [Aslam and Decatur, 1996, Cesa-Bianchi et al., 1999] that finiteness of the VC dimension characterizes learnability in the RCN model. Similarly, in the online mistake bound model, the parameter that characterizes learnability without noise — the Littestone dimension — continues to characterize learnability even in the presence of random label noise [Ben-David et al., 2009]. These results are for the so-called "0-1" loss. Learning with convex losses has been addressed only under limiting assumptions like separability or uniform noise rates [Manwani and Sastry, 2013].

In this paper, we consider risk minimization in the presence of *class-conditional* random label noise (abbreviated CCN). The data consists of iid samples from an underlying "clean" distribution $D$. The learning algorithm sees samples drawn from a noisy version $D_\rho$ of $D$ — where the noise rates depend on the class label. To the best of our knowledge, general results in this setting have not been obtained before. To this end, we develop two methods for suitably modifying *any given surrogate loss function* $\ell$, and show that minimizing the sample average of the modified proxy loss function

$\tilde{\ell}$ leads to provable risk bounds where the risk is calculated using the original loss $\ell$ on the clean distribution.

In our first approach, the modified or proxy loss is an unbiased estimate of the loss function. The idea of using unbiased estimators is well-known in stochastic optimization [Nemirovski et al., 2009], and regret bounds can be obtained for learning with noisy labels in an online learning setting (See Appendix B). Nonetheless, we bring out some important aspects of using unbiased estimators of loss functions for empirical risk minimization under CCN. In particular, we give a simple symmetry condition on the loss (enjoyed, for instance, by the Huber, logistic, and squared losses) to ensure that the proxy loss is also convex. Hinge loss does not satisfy the symmetry condition, and thus leads to a non-convex problem. We nonetheless provide a convex surrogate, leveraging the fact that the non-convex hinge problem is "close" to a convex problem (Theorem 6).

Our second approach is based on the fundamental observation that the minimizer of the risk (i.e. probability of misclassification) under the noisy distribution differs from that of the clean distribution *only* in where it thresholds $\eta(x) = P(Y = 1|x)$ to decide the label. In order to correct for the threshold, we then propose a simple weighted loss function, where the weights are label-dependent, as the proxy loss function. Our analysis builds on the notion of consistency of weighted loss functions studied by Scott [2012]. This approach leads to a very remarkable result that appropriately weighted losses like biased SVMs studied by Liu et al. [2003] are robust to CCN.

The main results and the contributions of the paper are summarized below:

1. To the best of our knowledge, we are the first to provide guarantees for risk minimization under random label noise in the general setting of convex surrogates, without any assumptions on the true distribution.
2. We provide two different approaches to suitably modifying any given surrogate loss function, that surprisingly lead to very similar risk bounds (Theorems 3 and 11). These general results include some existing results for random classification noise as special cases.
3. We resolve an elusive theoretical gap in the understanding of practical methods like biased SVM and weighted logistic regression — they are provably noise-tolerant (Theorem 11).
4. Our proxy losses are easy to compute — both the methods yield efficient algorithms.
5. Experiments on benchmark datasets show that the methods are robust even at high noise rates.

The outline of the paper is as follows. We introduce the problem setting and terminology in Section 2. In Section 3, we give our first main result concerning the method of unbiased estimators. In Section 4, we give our second and third main results for certain weighted loss functions. We present experimental results on synthetic and benchmark data sets in Section 5.

## 1.1 Related Work

Starting from the work of Bylander [1994], many noise tolerant versions of the perceptron algorithm have been developed. This includes the passive-aggressive family of algorithms [Crammer et al., 2006], confidence weighted learning [Dredze et al., 2008], AROW [Crammer et al., 2009] and the NHERD algorithm [Crammer and Lee, 2010]. The survey article by Khardon and Wachman [2007] provides an overview of some of this literature. A Bayesian approach to the problem of noisy labels is taken by Graepel and Herbrich [2000] and Lawrence and Schölkopf [2001]. As Adaboost is very sensitive to label noise, random label noise has also been considered in the context of boosting. Long and Servedio [2010] prove that any method based on a convex potential is inherently ill-suited to random label noise. Freund [2009] proposes a boosting algorithm based on a non-convex potential that is empirically seen to be robust against random label noise.

Stempfel and Ralaivola [2009] proposed the minimization of an unbiased proxy for the case of the hinge loss. However the hinge loss leads to a non-convex problem. Therefore, they proposed heuristic minimization approaches for which no theoretical guarantees were provided (We address the issue in Section 3.1). Cesa-Bianchi et al. [2011] focus on the online learning algorithms where they only need unbiased estimates of the gradient of the loss to provide guarantees for learning with noisy data. However, they consider a much harder noise model where *instances as well as labels* are noisy. Because of the harder noise model, they necessarily require multiple noisy copies per clean example and the unbiased estimation schemes also become fairly complicated. In particular, their techniques break down for non-smooth losses such as the hinge loss. In contrast, we show that unbiased estimation is always possible in the more benign random classification noise setting. Manwani and Sastry [2013] consider whether empirical risk minimization of the loss itself on the

noisy data is a good idea when the goal is to obtain small risk under the clean distribution. But it holds promise only for 0-1 and squared losses. Therefore, if empirical risk minimization over noisy samples has to work, we necessarily have to change the loss used to calculate the empirical risk. More recently, Scott et al. [2013] study the problem of classification under class-conditional noise model. However, they approach the problem from a different set of assumptions — the noise rates are *not* known, and the true distribution satisfies a certain "mutual irreducibility" property. Furthermore, they do not give any efficient algorithm for the problem.

## 2 Problem Setup and Background

Let $D$ be the underlying true distribution generating $(X, Y) \in \mathcal{X} \times \{\pm 1\}$ pairs from which $n$ iid samples $(X_1, Y_1), \ldots, (X_n, Y_n)$ are drawn. After injecting random classification noise (independently for each $i$) into these samples, corrupted samples $(X_1, \tilde{Y}_1), \ldots, (X_n, \tilde{Y}_n)$ are obtained. The class-conditional random noise model (CCN, for short) is given by:

$$P(\tilde{Y} = -1 | Y = +1) = \rho_{+1}, P(\tilde{Y} = +1 | Y = -1) = \rho_{-1}, \text{ and } \rho_{+1} + \rho_{-1} < 1$$

The corrupted samples are what the learning algorithm sees. We will assume that the noise rates $\rho_{+1}$ and $\rho_{-1}$ are known[1] to the learner. Let the distribution of $(X, \tilde{Y})$ be $D_\rho$. Instances are denoted by $\mathbf{x} \in \mathcal{X} \subseteq \mathbb{R}^d$. Noisy labels are denoted by $\tilde{y}$.

Let $f : \mathcal{X} \to \mathbb{R}$ be some real-valued decision function. The *risk* of $f$ w.r.t. the 0-1 loss is given by $R_D(f) = \mathbb{E}_{(X,Y) \sim D} \left[ 1_{\{\text{sign}(f(X)) \neq Y\}} \right]$. The optimal decision function (called Bayes optimal) that minimizes $R_D$ over all real-valued decision functions is given by $f^\star(x) = \text{sign}(\eta(x) - 1/2)$ where $\eta(x) = P(Y = 1|x)$. We denote by $R^*$ the corresponding *Bayes risk* under the clean distribution $D$, i.e. $R^* = R_D(f^*)$. Let $\ell(t, y)$ denote a loss function where $t \in \mathbb{R}$ is a real-valued prediction and $y \in \{\pm 1\}$ is a label. Let $\tilde{\ell}(t, \tilde{y})$ denote a suitably modified $\ell$ for use with noisy labels (obtained using methods in Sections 3 and 4). It is helpful to summarize the three important quantities associated with a decision function $f$:

1. Empirical $\tilde{\ell}$-risk on the observed sample: $\widehat{R}_{\tilde{\ell}}(f) := \frac{1}{n} \sum_{i=1}^{n} \tilde{\ell}(f(X_i), \tilde{Y}_i)$.

2. As $n$ grows, we expect $\widehat{R}_{\tilde{\ell}}(f)$ to be close to the $\tilde{\ell}$-risk under the noisy distribution $D_\rho$:

$$R_{\tilde{\ell}, D_\rho}(f) := \mathbb{E}_{(X, \tilde{Y}) \sim D_\rho} \left[ \tilde{\ell}(f(X), \tilde{Y}) \right] .$$

3. $\ell$-risk under the "clean" distribution $D$: $R_{\ell, D}(f) := \mathbb{E}_{(X,Y) \sim D} [\ell(f(X), Y)]$.

Typically, $\ell$ is a convex function that is *calibrated* with respect to an underlying loss function such as the 0-1 loss. $\ell$ is said to be *classification-calibrated* [Bartlett et al., 2006] if and only if there exists a convex, invertible, nondecreasing transformation $\psi_\ell$ (with $\psi_\ell(0) = 0$) such that $\psi_\ell(R_D(f) - R^*) \leq R_{\ell, D}(f) - \min_f R_{\ell, D}(f)$. The interpretation is that we can control the excess 0-1 risk by controlling the excess $\ell$-risk.

If $f$ is not quantified in a minimization, then it is implicit that the minimization is over all measurable functions. Though most of our results apply to a general function class $\mathcal{F}$, we instantiate $\mathcal{F}$ to be the set of hyperplanes of bounded $L_2$ norm, $\mathcal{W} = \{\mathbf{w} \in \mathbb{R}^d : \|\mathbf{w}\|_2 \leq W_2\}$ for certain specific results. Proofs are provided in the Appendix A.

## 3 Method of Unbiased Estimators

Let $\mathcal{F} : \mathcal{X} \to \mathbb{R}$ be a fixed class of real-valued decision functions, over which the empirical risk is minimized. The method of unbiased estimators uses the noise rates to construct an unbiased estimator $\tilde{\ell}(t, \tilde{y})$ for the loss $\ell(t, y)$. However, in the experiments we will tune the noise rate parameters through cross-validation. The following key lemma tells us how to construct unbiased estimators of the loss from noisy labels.

**Lemma 1.** *Let $\ell(t, y)$ be any bounded loss function. Then, if we define,*

$$\tilde{\ell}(t, y) := \frac{(1 - \rho_{-y}) \, \ell(t, y) - \rho_y \, \ell(t, -y)}{1 - \rho_{+1} - \rho_{-1}}$$

*we have, for any $t, y$,* $\quad \mathbb{E}_{\tilde{y}} \left[ \tilde{\ell}(t, \tilde{y}) \right] = \ell(t, y) .$

We can try to learn a good predictor in the presence of label noise by minimizing the sample average

$$\hat{f} \leftarrow \underset{f \in \mathcal{F}}{\operatorname{argmin}} \; \widehat{R}_{\tilde{\ell}}(f) \, .$$

By unbiasedness of $\tilde{\ell}$ (Lemma 1), we know that, for any fixed $f \in \mathcal{F}$, the above sample average converges to $R_{\ell,D}(f)$ even though the former is computed using noisy labels whereas the latter depends on the true labels. The following result gives a performance guarantee for this procedure in terms of the Rademacher complexity of the function class $\mathcal{F}$. The main idea in the proof is to use the contraction principle for Rademacher complexity to get rid of the dependence on the proxy loss $\tilde{\ell}$. The price to pay for this is $L_\rho$, the Lipschitz constant of $\tilde{\ell}$.

**Lemma 2.** *Let $\ell(t, y)$ be L-Lipschitz in t (for every y). Then, with probability at least $1 - \delta$,*

$$\max_{f \in \mathcal{F}} |\widehat{R}_{\tilde{\ell}}(f) - R_{\tilde{\ell}, D_\rho}(f)| \leq 2L_\rho \mathfrak{R}(\mathcal{F}) + \sqrt{\frac{\log(1/\delta)}{2n}}$$

*where $\mathfrak{R}(\mathcal{F}) := \mathbb{E}_{X_i, \epsilon_i} \left[ \sup_{f \in \mathcal{F}} \frac{1}{n} \epsilon_i f(X_i) \right]$ is the Rademacher complexity of the function class $\mathcal{F}$ and $L_\rho \leq 2L/(1 - \rho_{+1} - \rho_{-1})$ is the Lipschitz constant of $\tilde{\ell}$. Note that $\epsilon_i$'s are iid Rademacher (symmetric Bernoulli) random variables.*

The above lemma immediately leads to a performance bound for $\hat{f}$ with respect to the clean distribution $D$. Our first main result is stated in the theorem below.

**Theorem 3** (Main Result 1). *With probability at least $1 - \delta$,*

$$R_{\ell,D}(\hat{f}) \leq \min_{f \in \mathcal{F}} R_{\ell,D}(f) + 4L_\rho \mathfrak{R}(\mathcal{F}) + 2\sqrt{\frac{\log(1/\delta)}{2n}} \, .$$

*Furthermore, if $\ell$ is* classification-calibrated, *there exists a nondecreasing function $\zeta_\ell$ with $\zeta_\ell(0) = 0$ such that,*

$$R_D(\hat{f}) - R^* \leq \zeta_\ell \left( \min_{f \in \mathcal{F}} R_{\ell,D}(f) - \min_f R_{\ell,D}(f) + 4L_\rho \mathfrak{R}(\mathcal{F}) + 2\sqrt{\frac{\log(1/\delta)}{2n}} \right) \, .$$

The term on the right hand side involves both approximation error (that is small if $\mathcal{F}$ is large) and estimation error (that is small if $\mathcal{F}$ is small). However, by appropriately increasing the richness of the class $\mathcal{F}$ with sample size, we can ensure that the misclassification probability of $\hat{f}$ approaches the Bayes risk of the true distribution. This is despite the fact that the method of unbiased estimators computes the empirical minimizer $\hat{f}$ on a sample from the noisy distribution. Getting the optimal empirical minimizer $\hat{f}$ is efficient if $\tilde{\ell}$ is convex. Next, we address the issue of convexity of $\tilde{\ell}$.

### 3.1 Convex losses and their estimators

Note that the loss $\tilde{\ell}$ may not be convex even if we start with a convex $\ell$. An example is provided by the familiar hinge loss $\ell_{\text{hin}}(t, y) = [1 - yt]_+$. Stempfel and Ralaivola [2009] showed that $\tilde{\ell}_{\text{hin}}$ is not convex in general (of course, when $\rho_{+1} = \rho_{-1} = 0$, it is convex). Below we provide a simple condition to ensure convexity of $\tilde{\ell}$.

**Lemma 4.** *Suppose $\ell(t, y)$ is convex and twice differentiable almost everywhere in t (for every y) and also satisfies the symmetry property*

$$\forall t \in \mathbb{R}, \; \ell''(t, y) = \ell''(t, -y) \, .$$

*Then $\tilde{\ell}(t, y)$ is also convex in t.*

Examples satisfying the conditions of the lemma above are the squared loss $\ell_{\text{sq}}(t, y) = (t - y)^2$, the logistic loss $\ell_{\text{log}}(t, y) = \log(1 + \exp(-ty))$ and the Huber loss:

$$\ell_{\text{Hub}}(t, y) = \begin{cases} -4yt & \text{if } yt < -1 \\ (t - y)^2 & \text{if } -1 \leq yt \leq 1 \\ 0 & \text{if } yt > 1 \end{cases}$$

Consider the case where $\tilde{\ell}$ turns out to be non-convex when $\ell$ is convex, as in $\tilde{\ell}_{\text{hin}}$. In the online learning setting (where the adversary chooses a sequence of examples, and the prediction of a learner at round $i$ is based on the history of $i - 1$ examples with independently flipped labels), we could use a stochastic mirror descent type algorithm [Nemirovski et al., 2009] to arrive at risk bounds (See Appendix B) similar to Theorem 3. Then, we only need the expected loss to be convex and therefore

$\ell_{\text{hin}}$ does not present a problem. At first blush, it may appear that we do not have much hope of obtaining $\hat{f}$ in the iid setting efficiently. However, Lemma 2 provides a clue.

We will now focus on the function class $\mathcal{W}$ of hyperplanes. Even though $\widehat{R}_{\tilde{\ell}}(\mathbf{w})$ is non-convex, it is uniformly close to $R_{\tilde{\ell}, D_\rho}(\mathbf{w})$. Since $R_{\tilde{\ell}, D_\rho}(\mathbf{w}) = R_{\ell, D}(\mathbf{w})$, this shows that $\widehat{R}_{\tilde{\ell}}(\mathbf{w})$ is uniformly close to a convex function over $\mathbf{w} \in \mathcal{W}$. The following result shows that we can therefore approximately minimize $F(\mathbf{w}) = \widehat{R}_{\tilde{\ell}}(\mathbf{w})$ by minimizing the biconjugate $F^{\star\star}$. Recall that the (Fenchel) biconjugate $F^{\star\star}$ is the largest convex function that minorizes $F$.

**Lemma 5.** *Let $F : \mathcal{W} \to \mathbb{R}$ be a non-convex function defined on function class $\mathcal{W}$ such it is $\varepsilon$-close to a convex function $G : \mathcal{W} \to \mathbb{R}$:*
$$\forall \mathbf{w} \in \mathcal{W}, \ |F(\mathbf{w}) - G(\mathbf{w})| \leq \varepsilon$$
*Then any minimizer of $F^{\star\star}$ is a $2\varepsilon$-approximate (global) minimizer of $F$.*

Now, the following theorem establishes bounds for the case when $\tilde{\ell}$ is non-convex, via the solution obtained by minimizing the convex function $F^{**}$.

**Theorem 6.** *Let $\ell$ be a loss, such as the hinge loss, for which $\tilde{\ell}$ is non-convex. Let $\mathcal{W} = \{\mathbf{w} : \|\mathbf{w}_2\| \leq W_2\}$, let $\|X_i\|_2 \leq X_2$ almost surely, and let $\hat{\mathbf{w}}_{\text{approx}}$ be any (exact) minimizer of the convex problem*
$$\min_{\mathbf{w} \in \mathcal{W}} \ F^{\star\star}(\mathbf{w}) \,,$$
*where $F^{\star\star}(\mathbf{w})$ is the (Fenchel) biconjugate of the function $F(\mathbf{w}) = \widehat{R}_{\tilde{\ell}}(\mathbf{w})$. Then, with probability at least $1 - \delta$, $\hat{\mathbf{w}}_{\text{approx}}$ is a $2\varepsilon$-minimizer of $\widehat{R}_{\tilde{\ell}}(\cdot)$ where*
$$\varepsilon = \frac{2L_\rho X_2 W_2}{\sqrt{n}} + \sqrt{\frac{\log(1/\delta)}{2n}} \,.$$
*Therefore, with probability at least $1 - \delta$,*
$$R_{\ell, D}(\hat{\mathbf{w}}_{\text{approx}}) \leq \min_{\mathbf{w} \in \mathcal{W}} R_{\ell, D}(\mathbf{w}) + 4\varepsilon \,.$$

Numerical or symbolic computation of the biconjugate of a multidimensional function is difficult, in general, but can be done in special cases. It will be interesting to see if techniques from Computational Convex Analysis [Lucet, 2010] can be used to efficiently compute the biconjugate above.

## 4 Method of label-dependent costs

We develop the method of label-dependent costs from two key observations. First, the Bayes classifier for noisy distribution, denoted $\tilde{f}^*$, for the case $\rho_{+1} \neq \rho_{-1}$, simply uses a threshold different from $1/2$. Second, $\tilde{f}^*$ is the minimizer of a "label-dependent 0-1 loss" on the noisy distribution. The framework we develop here generalizes known results for the uniform noise rate setting $\rho_{+1} = \rho_{-1}$ and offers a more fundamental insight into the problem. The first observation is formalized in the lemma below.

**Lemma 7.** *Denote $P(Y = 1|X)$ by $\eta(X)$ and $P(\tilde{Y} = 1|X)$ by $\tilde{\eta}(X)$. The Bayes classifier under the noisy distribution, $\tilde{f}^* = \operatorname{argmin}_f E_{(X, \tilde{Y}) \sim D_\rho}\left[1_{\{\text{sign}(f(X)) \neq \tilde{Y}\}}\right]$ is given by,*
$$\tilde{f}^*(x) = \text{sign}(\tilde{\eta}(x) - 1/2) = \text{sign}\left(\eta(x) - \frac{1/2 - \rho_{-1}}{1 - \rho_{+1} - \rho_{-1}}\right).$$

Interestingly, this "noisy" Bayes classifier can also be obtained as the minimizer of a weighted 0-1 loss; which as we will show, allows us to "correct" for the threshold under the noisy distribution. Let us first introduce the notion of "label-dependent" costs for binary classification. We can write the 0-1 loss as a label-dependent loss as follows:
$$1_{\{\text{sign}(f(X)) \neq Y\}} = 1_{\{Y=1\}} 1_{\{f(X) \leq 0\}} + 1_{\{Y=-1\}} 1_{\{f(X) > 0\}}$$
We realize that the classical 0-1 loss is *unweighted*. Now, we could consider an $\alpha$-weighted version of the 0-1 loss as:
$$U_\alpha(t, y) = (1 - \alpha) 1_{\{y=1\}} 1_{\{t \leq 0\}} + \alpha 1_{\{y=-1\}} 1_{\{t > 0\}},$$
where $\alpha \in (0, 1)$. In fact we see that minimization w.r.t. the 0-1 loss is equivalent to that w.r.t. $U_{1/2}(f(X), Y)$. It is not a coincidence that Bayes optimal $f^*$ has a threshold $1/2$. The following lemma [Scott, 2012] shows that in fact for any $\alpha$-weighted 0-1 loss, the minimizer thresholds $\eta(x)$ at $\alpha$.

**Lemma 8** ($\alpha$-weighted Bayes optimal [Scott, 2012]). *Define $U_\alpha$-risk under distribution $D$ as*

$$R_{\alpha,D}(f) = E_{(X,Y)\sim D}[U_\alpha(f(X),Y)].$$

*Then, $f_\alpha^*(x) = sign(\eta(x) - \alpha)$ minimizes $U_\alpha$-risk.*

Now consider the risk of $f$ w.r.t. the $\alpha$-weighted 0-1 loss under noisy distribution $D_\rho$:

$$R_{\alpha,D_\rho}(f) = E_{(X,\tilde{Y})\sim D_\rho}\left[U_\alpha(f(X),\tilde{Y})\right].$$

At this juncture, we are interested in the following question: Does there exist an $\alpha \in (0,1)$ such that the minimizer of $U_\alpha$-risk under noisy distribution $D_\rho$ has the same sign as that of the Bayes optimal $f^*$? We now present our second main result in the following theorem that makes a stronger statement — the $U_\alpha$-risk under noisy distribution $D_\rho$ is linearly related to the 0-1 risk under the clean distribution $D$. The corollary of the theorem answers the question in the affirmative.

**Theorem 9** (Main Result 2). *For the choices,*

$$\alpha^* = \frac{1 - \rho_{+1} + \rho_{-1}}{2} \text{ and } A_\rho = \frac{1 - \rho_{+1} - \rho_{-1}}{2},$$

*there exists a constant $B_X$ that is independent of $f$ such that, for all functions $f$,*

$$R_{\alpha^*,D_\rho}(f) = A_\rho R_D(f) + B_X.$$

**Corollary 10.** *The $\alpha^\star$-weighted Bayes optimal classifier under noisy distribution coincides with that of 0-1 loss under clean distribution:*

$$\operatorname*{argmin}_f R_{\alpha^*,D_\rho}(f) = \operatorname*{argmin}_f R_D(f) = sign(\eta(x) - 1/2).$$

## 4.1 Proposed Proxy Surrogate Losses

Consider any surrogate loss function $\ell$; and the following decomposition:

$$\ell(t,y) = 1_{\{y=1\}}\ell_1(t) + 1_{\{y=-1\}}\ell_{-1}(t)$$

where $\ell_1$ and $\ell_{-1}$ are partial losses of $\ell$. Analogous to the 0-1 loss case, we can define $\alpha$-weighted loss function (Eqn. (1)) and the corresponding $\alpha$-weighted $\ell$-risk. Can we hope to minimize an $\alpha$-weighted $\ell$-risk with respect to noisy distribution $D_\rho$ and yet bound the excess 0-1 risk with respect to the clean distribution $D$? Indeed, the $\alpha^\star$ specified in Theorem 9 is precisely what we need. We are ready to state our third main result, which relies on a generalized notion of classification calibration for $\alpha$-weighted losses [Scott, 2012]:

**Theorem 11** (Main Result 3). *Consider the empirical risk minimization problem with noisy labels:*

$$\hat{f}_\alpha = \operatorname*{argmin}_{f\in\mathcal{F}} \frac{1}{n}\sum_{i=1}^n \ell_\alpha(f(X_i),\tilde{Y}_i).$$

*Define $\ell_\alpha$ as an $\alpha$-weighted margin loss function of the form:*

$$\ell_\alpha(t,y) = (1-\alpha)1_{\{y=1\}}\ell(t) + \alpha 1_{\{y=-1\}}\ell(-t) \tag{1}$$

*where $\ell : \mathbb{R} \to [0,\infty)$ is a convex loss function with Lipschitz constant $L$ such that it is classification-calibrated (i.e. $\ell'(0) < 0$). Then, for the choices $\alpha^*$ and $A_\rho$ in Theorem 9, there exists a nondecreasing function $\zeta_{\ell_{\alpha^\star}}$ with $\zeta_{\ell_{\alpha^\star}}(0) = 0$, such that the following bound holds with probability at least $1 - \delta$:*

$$R_D(\hat{f}_{\alpha^*}) - R^* \leq A_\rho^{-1}\zeta_{\ell_{\alpha^\star}}\left(\min_{f\in\mathcal{F}} R_{\alpha^*,D_\rho}(f) - \min_f R_{\alpha^*,D_\rho}(f) + 4L\mathfrak{R}(\mathcal{F}) + 2\sqrt{\frac{\log(1/\delta)}{2n}}\right).$$

Aside from bounding excess 0-1 risk under the clean distribution, the importance of the above theorem lies in the fact that it prescribes an efficient algorithm for empirical minimization with noisy labels: $\ell_\alpha$ is convex if $\ell$ is convex. Thus for any surrogate loss function including $\ell_{\text{hin}}$, $\hat{f}_{\alpha^*}$ can be efficiently computed using the method of label-dependent costs. Note that the choice of $\alpha^*$ above is quite intuitive. For instance, when $\rho_{-1} \ll \rho_{+1}$ (this occurs in settings such as Liu et al. [2003] where there are only positive and unlabeled examples), $\alpha^* < 1 - \alpha^*$ and therefore mistakes on positives are penalized more than those on negatives. This makes intuitive sense since an observed negative may well have been a positive but the other way around is unlikely. In practice we do not need to know $\alpha^*$, i.e. the noise rates $\rho_{+1}$ and $\rho_{-1}$. The optimization problem involves just one parameter that can be tuned by cross-validation (See Section 5).

# 5 Experiments

We show the robustness of the proposed algorithms to increasing rates of label noise on synthetic and real-world datasets. We compare the performance of the two proposed methods with state-of-the-art methods for dealing with random classification noise. We divide each dataset (randomly) into 3 training and test sets. We use a cross-validation set to tune the parameters specific to the algorithms. Accuracy of a classification algorithm is defined as the fraction of examples in the test set classified correctly *with respect to the clean distribution*. For given noise rates $\rho_{+1}$ and $\rho_{-1}$, labels of the training data are flipped accordingly and average accuracy over 3 train-test splits is computed[2]. For evaluation, we choose a representative algorithm based on each of the two proposed methods — $\tilde{\ell}_{\log}$ for the method of unbiased estimators and the widely-used C-SVM [Liu et al., 2003] method (which applies different costs on positives and negatives) for the method of label-dependent costs.

## 5.1 Synthetic data

First, we use the synthetic 2D linearly separable dataset shown in Figure 1(a). We observe from experiments that our methods achieve over 90% accuracy even when $\rho_{+1} = \rho_{-1} = 0.4$. Figure 1 shows the performance of $\tilde{\ell}_{\log}$ on the dataset for different noise rates. Next, we use a 2D UCI benchmark non-separable dataset ('banana'). The dataset and classification results using C-SVM (in fact, for uniform noise rates, $\alpha^* = 1/2$, so it is just the regular SVM) are shown in Figure 2. The results for higher noise rates are impressive as observed from Figures 2(d) and 2(e). The 'banana' dataset has been used in previous research on classification with noisy labels. In particular, the Random Projection classifier [Stempfel and Ralaivola, 2007] that learns a kernel perceptron in the presence of noisy labels achieves about 84% accuracy at $\rho_{+1} = \rho_{-1} = 0.3$ as observed from our experiments (as well as shown by Stempfel and Ralaivola [2007]), and the random hyperplane sampling method [Stempfel et al., 2007] gets about the same accuracy at $(\rho_{+1}, \rho_{-1}) = (0.2, 0.4)$ (as reported by Stempfel et al. [2007]). Contrast these with C-SVM that achieves about 90% accuracy at $\rho_{+1} = \rho_{-1} = 0.2$ and over 88% accuracy at $\rho_{+1} = \rho_{-1} = 0.4$.

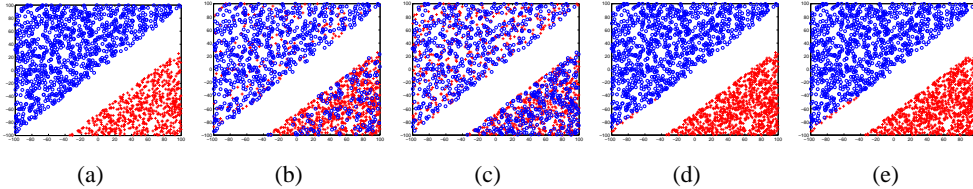

|        (a)        |        (b)        |        (c)        |        (d)        |        (e)        |

Figure 1: Classification of linearly separable synthetic data set using $\tilde{\ell}_{\log}$. The noise-free data is shown in the leftmost panel. Plots (b) and (c) show training data corrupted with noise rates ($\rho_{+1} = \rho_{-1} = \rho$) 0.2 and 0.4 respectively. Plots (d) and (e) show the corresponding classification results. The algorithm achieves 98.5% accuracy even at 0.4 noise rate per class. (Best viewed in color).

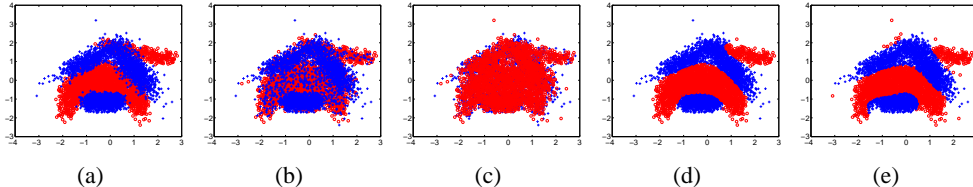

|        (a)        |        (b)        |        (c)        |        (d)        |        (e)        |

Figure 2: Classification of 'banana' data set using C-SVM. The noise-free data is shown in (a). Plots (b) and (c) show training data corrupted with noise rates ($\rho_{+1} = \rho_{-1} = \rho$) 0.2 and 0.4 respectively. Note that for $\rho_{+1} = \rho_{-1}$, $\alpha^* = 1/2$ (i.e. C-SVM reduces to regular SVM). Plots (d) and (e) show the corresponding classification results (Accuracies are 90.6% and 88.5% respectively). Even when 40% of the labels are corrupted ($\rho_{+1} = \rho_{-1} = 0.4$), the algorithm recovers the class structures as observed from plot (e). Note that the accuracy of the method at $\rho = 0$ is 90.8%.

## 5.2 Comparison with state-of-the-art methods on UCI benchmark

We compare our methods with three state-of-the-art methods for dealing with random classification noise: Random Projection (RP) classifier [Stempfel and Ralaivola, 2007]), NHERD

| DATASET $(d, n_+, n_-)$ | Noise rates | $\tilde{\ell}_{\log}$ | C-SVM | PAM | NHERD | RP |
|---|---|---|---|---|---|---|
| Breast cancer $(9, 77, 186)$ | $\rho_{+1} = \rho_{-1} = 0.2$ | **70.12** | 67.85 | **69.34** | 64.90 | **69.38** |
| | $\rho_{+1} = 0.3, \rho_{-1} = 0.1$ | **70.07** | 67.81 | 67.79 | 65.68 | 66.28 |
| | $\rho_{+1} = \rho_{-1} = 0.4$ | **67.79** | **67.79** | **67.05** | 56.50 | 54.19 |
| Diabetes $(8, 268, 500)$ | $\rho_{+1} = \rho_{-1} = 0.2$ | **76.04** | 66.41 | 69.53 | 73.18 | **75.00** |
| | $\rho_{+1} = 0.3, \rho_{-1} = 0.1$ | **75.52** | 66.41 | 65.89 | **74.74** | 67.71 |
| | $\rho_{+1} = \rho_{-1} = 0.4$ | 65.89 | 65.89 | 65.36 | **71.09** | 62.76 |
| Thyroid $(5, 65, 150)$ | $\rho_{+1} = \rho_{-1} = 0.2$ | 87.80 | 94.31 | **96.22** | 78.49 | 84.02 |
| | $\rho_{+1} = 0.3, \rho_{-1} = 0.1$ | 80.34 | **92.46** | 86.85 | 87.78 | 83.12 |
| | $\rho_{+1} = \rho_{-1} = 0.4$ | 83.10 | 66.32 | 70.98 | **85.95** | 57.96 |
| German $(20, 300, 700)$ | $\rho_{+1} = \rho_{-1} = 0.2$ | **71.80** | 68.40 | 63.80 | 67.80 | 62.80 |
| | $\rho_{+1} = 0.3, \rho_{-1} = 0.1$ | **71.40** | 68.40 | 67.80 | 67.80 | 67.40 |
| | $\rho_{+1} = \rho_{-1} = 0.4$ | 67.19 | **68.40** | 67.80 | 54.80 | 59.79 |
| Heart $(13, 120, 150)$ | $\rho_{+1} = \rho_{-1} = 0.2$ | **82.96** | 61.48 | 69.63 | **82.96** | 72.84 |
| | $\rho_{+1} = 0.3, \rho_{-1} = 0.1$ | **84.44** | 57.04 | 62.22 | 81.48 | 79.26 |
| | $\rho_{+1} = \rho_{-1} = 0.4$ | 57.04 | 54.81 | 53.33 | 52.59 | **68.15** |
| Image $(18, 1188, 898)$ | $\rho_{+1} = \rho_{-1} = 0.2$ | 82.45 | **91.95** | **92.90** | 77.76 | 65.29 |
| | $\rho_{+1} = 0.3, \rho_{-1} = 0.1$ | 82.55 | **89.26** | **89.55** | 79.39 | 70.66 |
| | $\rho_{+1} = \rho_{-1} = 0.4$ | 63.47 | 63.47 | **73.15** | 69.61 | 64.72 |

Table 1: Comparative study of classification algorithms on UCI benchmark datasets. Entries within 1% from the best in each row are in bold. All the methods except NHERD variants (which are not kernelizable) use Gaussian kernel with width 1. *All method-specific parameters are estimated through cross-validation.* Proposed methods ($\tilde{\ell}_{\log}$ and C-SVM) are competitive across all the datasets. We show the best performing NHERD variant ('project' and 'exact') in each case.

[Crammer and Lee, 2010]) (*project* and *exact* variants[3]), and perceptron algorithm with margin (PAM) which was shown to be robust to label noise by Khardon and Wachman [2007]. We use the standard UCI classification datasets, preprocessed and made available by Gunnar Rätsch(http://theoval.cmp.uea.ac.uk/matlab). For kernelized algorithms, we use Gaussian kernel with width set to the best width obtained by tuning it for a traditional SVM on the noise-free data. For $\tilde{\ell}_{\log}$, we use $\rho_{+1}$ and $\rho_{-1}$ that give the best accuracy in cross-validation. For C-SVM, we fix one of the weights to 1, and tune the other. Table 1 shows the performance of the methods for different settings of noise rates. C-SVM is competitive in 4 out of 6 datasets (Breast cancer, Thyroid, German and Image), while relatively poorer in the other two. On the other hand, $\tilde{\ell}_{\log}$ is competitive in all the data sets, and performs the best more often. When about 20% labels are corrupted, uniform ($\rho_{+1} = \rho_{-1} = 0.2$) and non-uniform cases ($\rho_{+1} = 0.3, \rho_{-1} = 0.1$) have similar accuracies in all the data sets, for both C-SVM and $\tilde{\ell}_{\log}$. Overall, we observe that the proposed methods are competitive and are able to tolerate moderate to high amounts of label noise in the data. Finally, in domains where noise rates are approximately known, our methods can benefit from the knowledge of noise rates. Our analysis shows that the methods are fairly robust to misspecification of noise rates (See Appendix C for results).

## 6 Conclusions and Future Work

We addressed the problem of risk minimization in the presence of random classification noise, and obtained general results in the setting using the methods of unbiased estimators and weighted loss functions. We have given efficient algorithms for both the methods with provable guarantees for learning under label noise. The proposed algorithms are easy to implement and the classification performance is impressive even at high noise rates and competitive with state-of-the-art methods on benchmark data. The algorithms already give a new family of methods that can be applied to the positive-unlabeled learning problem [Elkan and Noto, 2008], but the implications of the methods for this setting should be carefully analysed. We could consider harder noise models such as label noise depending on the example, and "nasty label noise" where labels to flip are chosen adversarially.

## 7 Acknowledgments

This research was supported by DOD Army grant W911NF-10-1-0529 to ID; PR acknowledges the support of ARO via W911NF-12-1-0390 and NSF via IIS-1149803, IIS-1320894.

## Footnotes

[1]This is not necessary in practice. See Section 5.

[2]Note that training and cross-validation are done on the noisy training data in our setting. To account for randomness in the flips to simulate a given noise rate, we repeat each experiment 3 times — independent corruptions of the data set for same setting of $\rho_{+1}$ and $\rho_{-1}$, and present the mean accuracy over the trials.

[3]A family of methods proposed by Crammer and coworkers [Crammer et al., 2006, 2009, Dredze et al., 2008] could be compared to, but [Crammer and Lee, 2010] show that the 2 NHERD variants perform the best.

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
