[Supplementary Material]

# A  Proofs

*Proof of Lemma 1.* One could directly compute and see that $\tilde{\ell}$ is unbiased. But to give a little more insight into what motivates the definition of $\tilde{\ell}$, consider the conditions that unbiasedness imposes on it. We should have, for every $t$,

$$\mathbb{E}_{\tilde{y} \overset{\rho}{\sim} y}\left[\tilde{\ell}(t, \tilde{y})\right] = \ell(t, y) .$$

Considering the cases $y = +1$ and $y = -1$ separately, gives the equations

$$(1 - \rho_{+1})\tilde{\ell}(t, +1) + \rho_{+1}\tilde{\ell}(t, -1) = \ell(t, +1) ,$$
$$(1 - \rho_{-1})\tilde{\ell}(t, -1) + \rho_{-1}\tilde{\ell}(t, +1) = \ell(t, -1) .$$

Solving these two equations for $\tilde{\ell}(t, +1)$ and $\tilde{\ell}(t, -1)$ gives

$$\tilde{\ell}(t, +1) = \frac{(1 - \rho_{-1})\ell(t, +1) - \rho_{+1}\ell(t, -1)}{1 - \rho_{+1} - \rho_{-1}} ,$$
$$\tilde{\ell}(t, -1) = \frac{(1 - \rho_{+1})\ell(t, -1) - \rho_{-1}\ell(t, +1)}{1 - \rho_{+1} - \rho_{-1}} .$$

$\square$

*Proof of Lemma 2.* By the basic Rademacher bound on the maximal deviation between risks and empirical risks over $f \in \mathcal{F}$, we get

$$\max_{f \in \mathcal{F}} |\widehat{R}_{\tilde{\ell}}(f) - R_{\tilde{\ell}, D_\rho}(f)| \leq 2 \cdot \mathfrak{R}(\tilde{\ell} \circ \mathcal{F}) + \sqrt{\frac{\log(1/\delta)}{2n}}$$

where

$$\mathfrak{R}(\tilde{\ell} \circ \mathcal{F}) := \mathbb{E}_{X_i, \tilde{Y}_i, \epsilon_i}\left[\sup_{f \in \mathcal{F}} \frac{1}{n} \sum_{i=1}^{n} \epsilon_i \tilde{\ell}(f(X_i), \tilde{Y}_i)\right]$$

If $\ell$ is $L$-Lipschitz then $\tilde{\ell}$ is $L_\rho$ Lipschitz for $L_\rho = (1 + |\rho_{+1} - \rho_{-1}|)L/(1 - \rho_{+1} - \rho_{-1}) \leq 2L/(1 - \rho_{+1} - \rho_{-1})$ and hence by the Lipschitz composition property of Rademacher averages, we have

$$\mathfrak{R}(\tilde{\ell} \circ \mathcal{F}) \leq L_\rho \cdot \mathfrak{R}(\mathcal{F}) .$$

$\square$

*Proof of Theorem 3.* Let $f^\star$ be the minimizer of $R_{\ell, D}(\cdot)$ over $\mathcal{F}$. We have

$$\begin{aligned}
&R_{\ell, D}(\hat{f}) - R_{\ell, D}(f^\star) \\
&= R_{\tilde{\ell}, D_\rho}(\hat{f}) - R_{\tilde{\ell}, D_\rho}(f^\star) \\
&= \widehat{R}_{\tilde{\ell}}(\hat{f}) - \widehat{R}_{\tilde{\ell}}(f^\star) + (R_{\tilde{\ell}, D_\rho}(\hat{f}) - \widehat{R}_{\tilde{\ell}}(\hat{f})) \\
&\quad + (\widehat{R}_{\tilde{\ell}}(f^\star) - R_{\tilde{\ell}, D_\rho}(f^\star)) \\
&\leq 0 + 2 \max_{f \in \mathcal{F}} |\widehat{R}_{\tilde{\ell}}(f) - R_{\tilde{\ell}, D_\rho}(f)| .
\end{aligned}$$

We can now apply Lemma 2 to control the last quantity above, and thus obtain the first statement of the theorem. Now, if $\ell$ is *classification-calibrated*, then from Theorem 1 of [Bartlett et al., 2006], we know there exists a convex, invertible, nondecreasing transformation $\psi_\ell$ with $\psi_\ell(0) = 0$ such that,

$$\psi_\ell(R_D(f) - R^*) \leq R_{\ell, D}(f) - \inf_f R_{\ell, D}(f)$$

Subtracting $\min_f R_{\ell, D}(f)$ off either sides of the first inequality in the theorem statement, and realizing that $\psi_\ell^{-1}$ is nondecreasing as well, with $\psi_\ell^{-1}(0) = 0$, we conclude:

$$R_D(\hat{f}) - R^* \leq \psi_\ell^{-1}\left(\min_{f \in \mathcal{F}} R_{\ell, D}(f) - \min_f R_{\ell, D}(f) + 4L_\rho\mathfrak{R}(\mathcal{F}) + 2\sqrt{\frac{\log(1/\delta)}{2n}}\right) .$$

$\square$

*Proof of Lemma 4.* Let us compute $\tilde{\ell}''(t, y)$ (recall that differentiation is w.r.t. $t$) and show that it is non-negative under the symmetry condition $\ell''(t, y) = \ell''(t, -y)$. We have

$$
\begin{aligned}
\tilde{\ell}''(t, y) &= \frac{(1 - \rho_{-y})\ell''(t, y) - \rho_y \ell''(t, -y)}{1 - \rho_{+1} - \rho_{-1}} \\
&= \frac{(1 - \rho_{-y})\ell''(t, y) - \rho_y \ell''(t, y)}{1 - \rho_{+1} - \rho_{-1}} \\
&= \frac{(1 - \rho_{-y} - \rho_y)\ell''(t, y)}{1 - \rho_{+1} - \rho_{-1}} \\
&= \ell''(t, y) \geq 0 \,,
\end{aligned}
$$

since $\ell$ is convex in $t$. $\qquad\square$

*Proof of Lemma 5.* Since $F \geq G - \varepsilon$ and $F^{\star\star}$ is the largest convex function that minorizes $F$, we must have $F^{\star\star} \geq G - \varepsilon$. This means that $F^{\star\star} + 2\varepsilon \geq G + \varepsilon \geq F$. Thus, $F$ is sandwiched between $F^{\star\star} + 2\varepsilon$ and $F^{\star\star}$. The lemma follows directly from this. $\qquad\square$

*Proof of Theorem 6.* The first part of the theorem follows by combining Lemma 2 and Lemma 5, using the fact that if $\|\mathbf{w}\|_2 \leq W_2$ for any $\mathbf{w}$ and $\|X_i\|_2 \leq X_2$ then, $\mathfrak{R}(\mathcal{W}) \leq W_2 X_2 / \sqrt{n}$. The second part follows by noting that Theorem 3 is true also for $2\varepsilon$-minimizers of the empirical risk $\widehat{R}_{\tilde{\ell}}$ provided we add $2\varepsilon$ to the right hand side. $\qquad\square$

*Proof of Lemma 7.* The first equality is true because the optimal bayes classifier under $D_\rho$ thresholds $\tilde{\eta}(X) = P(\tilde{Y} = 1|X)$ at 1/2. Now,

$$
\begin{aligned}
\tilde{\eta}(X) &= P(\tilde{Y} = 1, Y = 1|X) + P(\tilde{Y} = 1, Y = -1|X) \\
&= P(\tilde{Y} = 1|Y = 1)P(Y = 1|X) + P(\tilde{Y} = 1|Y = -1)P(Y = -1|X) \\
&= (1 - \rho_{+1})\eta(X) + \rho_{-1}(1 - \eta(X)) \\
&= (1 - \rho_{+1} - \rho_{-1})\eta(X) + \rho_{-1}.
\end{aligned}
$$

Therefore,

$$
\begin{aligned}
\operatorname{sign}(\tilde{\eta}(x) - 1/2) &= \operatorname{sign}((1 - \rho_{+1} - \rho_{-1})\eta(x) + \rho_{-1} - 1/2) \\
&= \operatorname{sign}\left(\eta(x) - \frac{1/2 - \rho_{-1}}{1 - \rho_{+1} - \rho_{-1}}\right).
\end{aligned}
$$

$\qquad\square$

*Proof of Theorem 9.* Let us think of $f$ as $\{\pm 1\}$-valued since both $C_D$ and $C_{\alpha, D_\rho}$ depend only on $\operatorname{sign}(f)$. We have,

$$
C_D(f) = \mathbb{E}_Y \left[ 1_{\{f(X) \neq Y\}} \right]
$$

and

$$
C_{\alpha, D_\rho}(f) = \mathbb{E}_{\tilde{Y}} \left[ (1 - \alpha)1_{\{\tilde{Y}=1\}} 1_{\{f(X) \neq 1\}} + \alpha 1_{\{\tilde{Y}=-1\}} 1_{\{f(X) \neq -1\}} \right].
$$

Note that $R_D(f) = \mathbb{E}_X [C_D(f)]$, and $R_{\alpha, D_\rho}(f) = \mathbb{E}_X [C_{\alpha, D_\rho}(f)]$. Also note that $C_D(f) = \eta(X)$ if $f(X) = -1$, and $C_D(f) = 1 - \eta(X)$ otherwise.
Similarly, $C_{\alpha, D_\rho}(f) = (1 - \alpha)\tilde{\eta}(X)$ if $f(X) = -1$ and $C_{\alpha, D_\rho}(f) = \alpha(1 - \tilde{\eta}(X))$ otherwise. We want to find $A$ and $B$ such that the following equations hold simultaneously:

$$
\begin{aligned}
(1 - \alpha)\tilde{\eta}(X) &= A\eta(X) + B \\
\alpha(1 - \tilde{\eta}(X)) &= A(1 - \eta(X)) + B
\end{aligned}
$$

Using the relation between $\eta(X)$ and $\tilde{\eta}(X)$ in Lemma 7 and solving for $A$ we get,

$$
A = \frac{(1 - \rho_{+1} - \rho_{-1})\eta(X) + \rho_{-1} - \alpha}{2\eta(X) - 1}.
$$

Choosing $\alpha = \alpha^* = \frac{1-\rho_{+1}+\rho_{-1}}{2}$, and simplifying, we get a constant $A$ that depends only on the noise rates:

$$A = A_\rho = \frac{1-\rho_{+1}-\rho_{-1}}{2}.$$

Consequently,

$$B = \rho_{-1}(1-\alpha^*) - \frac{\alpha^*}{2}(1-\rho_{+1}-\rho_{-1})\eta(X).$$

Taking expectation with respect to $X$, we conclude:

$$R_{\alpha^*,D_\eta}(f) = A_\rho R_D(f) + B_X,$$

where $B_X = \mathbb{E}_X[B]$.
$\qquad\qquad\qquad\qquad\qquad\qquad\qquad\qquad\qquad\qquad\qquad\qquad\qquad\qquad\quad\square$

*Proof of Corollary 10.* The proof is immediate from Theorem 9 observing that $B_X$ is independent of $f$.
$\qquad\qquad\qquad\qquad\qquad\qquad\qquad\qquad\qquad\qquad\qquad\qquad\qquad\qquad\qquad\quad\square$

*Proof of Theorem 11.* From Corollary 4.1 in [Scott, 2012], we can infer that $\ell_\alpha$ is $\alpha$-CC for given $\alpha \in (0,1)$, as $\ell$ is convex, classification-calibrated and $\ell'(0) < 0$. Then, from Theorem 3.1 in [Scott, 2012], there exists an *invertible*, *non-decreasing* convex transformation $\psi_{\ell_\alpha}$ with $\psi_{\ell_\alpha}(0) = 0$ such that, for any $f$ and any distribution $D$,

$$\psi_{\ell_\alpha}(R_{\alpha,D}(f) - \min_f R_{\alpha,D}(f)) \leq R_{\ell_\alpha,D}(f) - \min_f R_{\ell_\alpha,D}(f).$$

Fix distribution to be $D_\rho$, and let $f = \hat{f}_\alpha$. The RHS of the above inequality can then be controlled similarly as in the proof of Theorem 3. It is easy to see that the Lipschitz constant of $\ell_\alpha$ is same as that of $\ell$, denoted $L$. With probability at least $1 - \delta$:

$$R_{\ell_\alpha,D_\rho}(\hat{f}_\alpha) - \min_{f\in\mathcal{F}} R_{\ell_\alpha,D_\rho}(f) \leq 4L\mathfrak{R}(\mathcal{F}) + 2\sqrt{\frac{\log(1/\delta)}{2n}}.$$

Now consider $R_{\alpha,D_\rho}(f) - \min_f R_{\alpha,D_\rho}(f)$. Using the linear relationship between $R_{\alpha,D_\rho}$ and $R_D$ at $\alpha^*$ (Theorem 9), we get $R_{\alpha^*,D_\rho}(f) - \min_f R_{\alpha^*,D_\rho}(f) = A_\rho(R_D(f) - R^*)$. $B_X$ vanishes because it is constant for the distribution $D_\rho$. Note that $\psi_{\ell_{\alpha^*}}^{-1}$ is nondecreasing as well and $\psi_{\ell_{\alpha^*}}^{-1}(0) = 0$. Subtracting $\min_f R_{\alpha^*,D_\rho}(f)$ from both sides of the second inequality above, the statement of the theorem follows: With probability at least $1 - \delta$,

$$R_D(\hat{f}_{\alpha^*}) - R^* \leq A_\rho^{-1}\psi_{\ell_{\alpha^\star}}^{-1}\left(\min_{f\in\mathcal{F}} R_{\alpha^*,D_\rho}(f) - \min_f R_{\alpha^*,D_\rho}(f) + 4L\mathfrak{R}(\mathcal{F}) + 2\sqrt{\frac{\log(1/\delta)}{2n}}\right).$$

$\qquad\qquad\qquad\qquad\qquad\qquad\qquad\qquad\qquad\qquad\qquad\qquad\qquad\qquad\qquad\quad\square$

## B  Online learning

Consider the setting where an adversary chooses a sequence $(\mathbf{x}_1, y_1), \ldots, (\mathbf{x}_n, y_n)$ of examples. At time $i$, the learner has to make a prediction based on $(\mathbf{x}_1, \tilde{y}_1), \ldots, (\mathbf{x}_{i-1}, \tilde{y}_{i-1})$ and $\mathbf{x}_i$ where $\tilde{y}_i$ are the noisy labels. But the learner's cumulative loss as well as that of the best fixed predictor in hindsight are both computed using the true labels $y_i$. Note that if $\ell(t, y)$ is convex in $t$ (for every $y$), and we choose $\lambda_1 \in \partial\ell(t, y)$ and $\lambda_2 \in \partial\ell(t, -y)$, (where $\partial\ell$ is the subdifferential w.r.t. $t$) we have

$$\mathbb{E}_{\tilde{y}}[g(t, \tilde{y})] \in \partial\ell(t, y) \qquad\qquad\qquad\qquad (2)$$

where

$$g(t, y) = \frac{(1-\rho_{-y})\lambda_1 - \rho_y\lambda_2}{1-\rho_{+1}-\rho_{-1}} \qquad\qquad\qquad\qquad (3)$$

We show that Algorithm 1 indeed satisfies low regret (in expectation) on the original sequence chosen by the adversary even though it only receives noisy versions of the labels. We fix the function class to be the set $\mathcal{W}$ of bounded-norm hyperplanes.

---
**Algorithm 1** Online learning using unbiased gradients
---
Choose learning rate $\gamma > 0$
$\mathcal{W} = \{\mathbf{w} \ : \ \|\mathbf{w}\|_2 \leq W_2\}$
$\Pi_\mathcal{W}(\cdot) =$ Euclidean projection onto $\mathcal{W}$
Initialize $\mathbf{w}_0 \leftarrow \mathbf{0}$
**for** $i = 1$ to $n$ **do**
    Receive $\mathbf{x}_i \in \mathbb{R}^d$
    Predict $\langle \mathbf{w}_{i-1}, \mathbf{x}_i \rangle$
    Receive noisy label $\tilde{y}_i$
    Update $\mathbf{w}_i \leftarrow \Pi_\mathcal{W}(\mathbf{w}_{i-1} - \gamma g(\langle \mathbf{w}_{i-1}, \mathbf{x}_i \rangle, \tilde{y}_i)\mathbf{x}_i)$ where $g(\cdot, \cdot)$ is defined in (3)
**end for**
---

**Theorem 12.** *Let $\ell(t,y)$ be convex and $L$-Lipschitz in $t$ (for every $y$). Fix an arbitrary sequence $(\mathbf{x}_1, y_1), \ldots, (\mathbf{x}_n, y_n)$. If Algorithm 1 is run on noisy data set $(\mathbf{x}_1, \tilde{y}_1), \ldots, (\mathbf{x}_n, \tilde{y}_n)$ with learning rate $\gamma = W_2/(X_2 L_\rho \sqrt{n})$ where $\tilde{y}_i$ is noisy version of $y_i$ with noise rates $\rho_{+1}, \rho_{-1}$, then we have*

$$\mathbb{E}_{\tilde{y}_{1:n}} \left[ \max_{\|\mathbf{w}\|_2 \leq W_2} \sum_{i=1}^n (\ell(\langle \mathbf{w}_{i-1}, \mathbf{x}_i \rangle, y_i) - \ell(\langle \mathbf{w}, \mathbf{x}_i \rangle, y_i)) \right] \leq L_\rho X_2 W_2 \sqrt{n} \,,$$

*where $L_\rho := (1 + |\rho_{+1} - \rho_{-1}|)L/(1 - \rho_{+1} - \rho_{-1})$ and it is assumed that $\|\mathbf{x}_i\| \leq X_2$ for all $i \in [n]$.*

*Proof.* Let us use the abbreviation $g_i$ for $g(\langle \mathbf{w}_{i-1}, \mathbf{x}_i \rangle, \tilde{y}_i)\mathbf{x}_i$ so that the update in Algorithm 1 becomes $\mathbf{w}_i \leftarrow \Pi_\mathcal{W}(\mathbf{w}_{i-1} - \gamma g_i)$. It is well known [Zinkevich, 2003] that, for any $\mathbf{w}$,

$$\sum_{i=1}^n \langle g_i, \mathbf{w}_{i-1} - \mathbf{w} \rangle \leq \frac{\gamma}{2} \sum_{i=1}^n \|g_i\|^2 + \frac{\|\mathbf{w}\|^2}{2\gamma} \,. \tag{4}$$

Since $\ell$ is $L$-Lipschitz, the $\lambda_1, \lambda_2$ appearing in the definition (3) of $g(\cdot, \cdot)$ satisfy $|\lambda_1|, |\lambda_2| \leq L$. This implies $|g(t,y)| \leq (1 + |\rho_{+1} - \rho_{-1}|)L/(1 - \rho_{+1} - \rho_{-1}) = L_\rho$ and hence $\|g_i\| \leq L_\rho X_2$. Thus, we have, for any $\mathbf{w}$ with $\|\mathbf{w}\| \leq W_2$, $\sum_{i=1}^n \langle g_i, \mathbf{w}_{i-1} - \mathbf{w} \rangle \leq \frac{\gamma L_\rho^2 X_2^2 n}{2} + \frac{W_2^2}{2\gamma}$. Choosing $\gamma = (W_2/L_\rho X_2)\frac{1}{\sqrt{n}}$, we get $\sum_{i=1}^n \langle g_i, \mathbf{w}_{i-1} - \mathbf{w} \rangle \leq L_\rho X_2 W_2 \sqrt{n}$. Note that $\mathbf{w}_{i-1}$ only depends on $\tilde{y}_{1:i-1}$. Hence

$$\mathbb{E}_{\tilde{y}_i} \left[ \langle g_i, \mathbf{w}_{i-1} - \mathbf{w} \rangle \mid \tilde{y}_{1:i-1} \right] = \langle \mathbb{E}_{\tilde{y}_i} \left[ g_i \mid \tilde{y}_{1:i-1} \right], \mathbf{w}_{i-1} - \mathbf{w} \rangle \geq \ell(\langle \mathbf{w}_{i-1}, \mathbf{x}_i \rangle, y_i) - \ell(\langle \mathbf{w}, \mathbf{x}_i \rangle, y_i)$$

because $\mathbb{E}_{\tilde{y}_i} \left[ g_i \mid \tilde{y}_{1:i-1} \right] \in \partial_{\mathbf{w}=\mathbf{w}_{i-1}} \ell(\langle \mathbf{w}, \mathbf{x}_i \rangle, y_i)$ by (2) and the chain rule for differentiation, and $\ell(\langle \mathbf{w}, \mathbf{x}_i \rangle, y_i)$ is convex in $\mathbf{w}$. Thus, for any $\mathbf{w}$ with $\|\mathbf{w}\|_2 \leq W_2$,

$$\mathbb{E}_{\tilde{y}_{1:n}} \left[ \sum_{i=1}^n \ell(\langle \mathbf{w}_{i-1}, \mathbf{x}_i \rangle, y_i) \right] - \sum_{i=1}^n \ell(\langle \mathbf{w}, \mathbf{x}_i \rangle, y_i) \leq L_\rho X_2 W_2 \sqrt{n}.$$

Since the above inequality is true for any $\mathbf{w}$ with $\|\mathbf{w}\|_2 \leq 1$, we have

$$\mathbb{E}_{\tilde{y}_{1:n}} \left[ \sum_{i=1}^n \ell(\langle \mathbf{w}_{i-1}, \mathbf{x}_i \rangle, y_i) \right] - \min_{\|\mathbf{w}\|_2 \leq W_2} \sum_{i=1}^n \ell(\langle \mathbf{w}, \mathbf{x}_i \rangle, y_i) \leq L_\rho X_2 W_2 \sqrt{n}.$$

Observing that the minimum over $\mathbf{w}$ is not random allows us to move it inside the expectation giving us the theorem. $\qquad\square$

## C  Experiments

### C.1  Knowledge of noise rates

The proposed algorithms require the knowledge of noise rates $\rho_{+1}$ and $\rho_{-1}$. However, in practice, we do not know the true value of noise rates, and therefore we resort to cross-validating the values in our experiments. We emphasize here that in case the true noise rates are known, our methods can benefit from that knowledge as observed from our experiments (results not shown), whereas the

Figure 3: Study of sensitivity of batch ($\tilde{\ell}_{\log}$) and online (Hinge, Huber and Logistic) methods (Algorithm 1) to specification of noise rates $\rho_{+1}$ and $\rho_{-1}$. True noise rates $\rho_{+1} = \rho_{-1} = \rho$ are misspecified as $(\rho_{+1} \pm \epsilon, \rho_{-1} \pm \epsilon)$ for $\epsilon \in \{0.1, 0.2, 0.3, 0.4\}$. The ratio between the average accuracy for a given $\epsilon$ and the accuracy at $\epsilon = 0$, i.e. when true noise rates are specified, is plotted for different values of noise rates $\rho$. The ratio is computed for each of the 6 UCI data sets in Table 1 and the mean and the standard deviation of the ratios are shown. Ratio being equal to 1 for a given $\epsilon$ means that the performance of the algorithm, on average, is unaltered by misspecification of noise rates up to $\epsilon$. As expected, the ratio decreases, i.e. the algorithms perform worse as $\epsilon$ increases. Most of the ratios being close to 1 suggests that the proposed methods are fairly robust with respect to $\epsilon$-misspecification of noise rates.

competitive methods *cannot* as they do not involve noise rates. In some cases (and domains), we may be able to approximately specify noise rates. This motivates our study presented in Figure 3. True noise rates $\rho_{+1} = \rho_{-1} = \rho$ are misspecified as $(\rho_{+1} \pm \epsilon, \rho_{-1} \pm \epsilon)$ for $\epsilon \in \{0.1, 0.2, 0.3, 0.4\}$. The ratio between the average accuracy for a given $\epsilon$ and the accuracy at $\epsilon = 0$, i.e. when true noise rates are specified, is a measure of sensitivity of the algorithms to $\epsilon$-misspecification of noise rates. We would want the ratio to be close to 1 for a given $\epsilon$, which would suggest that the method is fairly robust with respect to the $\epsilon$-misspecification. The results in Figure 3 show that the proposed methods are robust to $\epsilon$-misspecification of noise rates, which in turn suggests that our methods can find better use in applications where labels can be noisy *and* noise rates are approximately known, without resorting to ad-hoc cross-validation procedures on the noisy data.