[Reviews · NeurIPS 2013]

Submitted by Assigned_Reviewer_6

The authors study the problem of binary classification in the presence of random, class-conditional noise in the training data. They propose two approaches based on a suitable modification of a given surrogate loss function and derive performance bounds. More specifically, they provide guarantees for risk minimization of convex surrogates under random label noise in the general setting, and without any assumptions on the true distribution. Moreover, they provide two alternative approaches for modifying a given surrogate loss function. Experiments are presented on synthetic data and some benchmark datasets.

The paper is well-written and appears to be technically sound. The theoretical results are non-trivial but not extremely profound either, especially in light of existing work on related setting.

What could be called into question is the relevance of the setting. Is the assumption of having constant class noise in the training data indeed a realistic one? This question appears to be legitimate despite the theoretical nature of the paper.
Summary: The paper is well-written and appears to be technically sound. The theoretical results are non-trivial but not very profound either, especially in light of existing work on related setting. What could be called into question is the relevance of the setting.

Submitted by Assigned_Reviewer_7

Learning with Noisy Labels

The paper addresses the problem of binary classification in the situation
where the training labels are corrupted by class-conditional random noise.
The authors propose 2 surrogate-loss based learning methods to address the problem:
the first exploits a simple symmetry condition on the loss function used to
provide a simple unbiased estimator of the non-noisy risk and the second
promotes the use of a weighted 0-1 loss that comes from an appropriate reduction
of the problem of learning from noisy labels.

The paper is a very clean and strong contribution. It provides original
theoretical results (e.g. learnability with convex surrogate in the case
of noise, noise-tolerance of SVMs, and so on) as well as compelling empirical
results. Everything is wrapped up in a nicely written paper.

I essentially have questions on future directions:
- the authors mention adversarial noise: before going to this point, is there something that can be said
about learnability with monotonic noise, as defined by Bylander ?
- what about richer noise models like Constant Partition Classification Noise (CPCN) noise ?
Summary: Very good paper, providing significant result to learn binary classifiers from noisy labels using convex surrogates. Technical results are important, the writing is good and the experiments are compelling.

Submitted by Assigned_Reviewer_8

Overall, I found the paper quite well-written.

I think the assertion in the abstract (and then again in the Experiments
section), that you learn with 88% accuracy when 80% labels are flipped is
incorrect. You are only flipping 40% of the labels (the probability of
flip is convex combination of \rho_+ and \rho_- and not their sum). In
any, case the goal is to get better accuracy as noise rate goes to half,
not one (where the problem is trivially easy with just all labels
flipped).

I wonder if it is worthwhile to compare your methods with Kearns' SQ
setting. Since minimizing a convex loss can be done by gradient methods
(which have statistical analogues). And so you would get tolerance to
random classification noise for free. (Kearns does not allow
class-conditional noise, but I think that part can be handled easily.) I'm
not sure what kind of excess risk bounds you would get by such
SQ-simulation.

Minor quibbles:
--------------
1. You use the term PU learning, without ever defining it.
2. You use "so-called" very often. Especially for zero-one loss. Why is it
so-called? If you have objection to the name you should state it.

---
Update: Regarding using the SQ model. Even if you are using surrogate loss, your optimization problem can be solved using an algorithm that only makes statistical queries rather than data points.
Summary: This paper considers the problem of learning in the presence of random
classification noise. In contrast with the PAC-like models, the main
goal here is if the goal is to minimize some convex loss function (with
respect to the true models), this can be done by suitable modifications
even when the labels are noisy (in many cases).

The paper also contains some experiments studying the proposed methods and
other related techniques.
Author Feedback

Author rebuttal: ---Assigned_Reviewer_6---
We are not claiming that training data generated / obtained in real-world tend to have constant class noise. The noise model (CCN) has already been studied in the literature. While some results are known under certain assumptions on the data distribution; it is quite surprising that even under the simple noise model, no guarantees are known for minimizing surrogate losses with noisy training data, especially given that almost two decades have elapsed since the first work on classification with random noise. The paper does not only serve to settle the question in the affirmative, but also helps advance our understanding of known algorithms such as biased SVMs.

---Assigned_Reviewer_7---
We thank the reviewer for the encouraging comments and very pertinent thoughts for further research. The CPCN model assumes that the example space is partitioned such that each partition has a noise rate. Recent work by Ralaivola et. al establishes that if a function is PAC-learnable in the uniform noise rate model (CN), it is also PAC-learnable under CPCN (thus showing that the models are equivalent under PAC-learning). It would be interesting to see if our CCN learning algorithms can be cleverly used to learn under CPCN model, akin to the PAC-learnability argument for CCN=CPCN. Indeed, studying learnability under CPCN model seems to be the right question to ask given our current results. On the other hand, the monotonic noise model defined by Bylander, where the noise rate decreases monotonically with distance of example from the target hyperplane, is more realistic. CPCN model, in some sense, is a discrete version (albeit without monotonicity) of the monotonic noise model, and understanding either of the two models better would be a step forward.

---Assigned_Reviewer_8---
Thanks to the reviewer for pointing this out --- only 40% of the labels are flipped and not 80%. The idea of casting gradient based convex minimization algorithms in Kearns' Statistical Query (SQ) framework to obtain noise-tolerant algorithms 'for free' is an intriguing one and does seem worth pursuing. However, the focus of this work is on modifying *surrogates* to obtain noise-tolerant algorithms. The SQ technique, if successful, would work by modifying *algorithms* (ensuring, for instance, that they only make statistical queries). Also, we use "so-called" only to mean "commonly known as", but the reviewer's question shows the overlooked ambiguity of the usage -- we will remove this ambiguity in future versions.